# Hyperoxia Reprogrammes Microvascular Endothelial Cell Response to Hypoxia in an Organ-Specific Manner

**DOI:** 10.3390/cells11162469

**Published:** 2022-08-09

**Authors:** Moritz Reiterer, Amanda Eakin, Randall S. Johnson, Cristina M. Branco

**Affiliations:** 1Patrick G Johnston Centre for Cancer Research, Queen’s University, Belfast BT9 7AE, UK; 2Department of Physiology, Development and Neuroscience, University of Cambridge, Cambridge CB2 3EG, UK

**Keywords:** microvascular endothelium, heterogeneity, hyperoxia, hypoxia, metabolism

## Abstract

Organ function relies on microvascular networks to maintain homeostatic equilibrium, which varies widely in different organs and during different physiological challenges. The endothelium role in this critical process can only be evaluated in physiologically relevant contexts. Comparing the responses to oxygen flux in primary murine microvascular EC (MVEC) obtained from brain and lung tissue reveals that supra-physiological oxygen tensions can compromise MVEC viability. Brain MVEC lose mitochondrial activity and undergo significant alterations in electron transport chain (ETC) composition when cultured under standard, non-physiological atmospheric oxygen levels. While glycolytic capacity of both lung and brain MVEC are unchanged by environmental oxygen, the ability to trigger a metabolic shift when oxygen levels drop is greatly compromised following exposure to hyperoxia. This is particularly striking in MVEC from the brain. This work demonstrates that the unique metabolism and function of organ-specific MVEC (1) can be reprogrammed by external oxygen, (2) that this reprogramming can compromise MVEC survival and, importantly, (3) that ex vivo modelling of endothelial function is significantly affected by culture conditions. It further demonstrates that physiological, metabolic and functional studies performed in non-physiological environments do not represent cell function in situ, and this has serious implications in the interpretation of cell-based pre-clinical models.

## 1. Introduction

Microvascular endothelial cells (MVEC) are the most heterogeneous, versatile and specialised cells within the vascular tree [1,2]. They are essential for the local regulation of nutrient delivery, signalling and gas exchange rates to meet organ demand in tune with systemic availability [3,4]. Angiocrine stimuli from resident MVEC mediate organ function and tissue microenvironment are highly diverse in complex organisms [5,6,7]. Oxygen availability is dynamic and dependent on a tissue’s cellular components and metabolic activity and, as such, the perception and response to oscillations in oxygen levels by MVEC is critical to organ homeostasis and effectively sidesteps autonomic control [3,5]. This plasticity underlies functions beyond perfusion and permeability, and ensures organ performance in a variety of physiological contexts, such as exercise or high altitude [8,9], multiple pathologies [10,11,12,13] and during tissue remodelling and regeneration [6,7,14]. MVEC are functionally and metabolically equipped to respond to hypoxia, and organisms are, for the most part, capable of seamlessly adjusting to O_2_ flux at systemic and local levels. When stretched too far or for too long beyond the normal physiological range, these cellular responses can exacerbate pathologies, typically involving vascular dysfunction [3], as seen in diabetic retinopathy [15] or atherosclerosis [16].

MVEC can also be exposed to supra-physiological oxygen levels or hyperoxia. This is experienced when O_2_ levels are significantly above those typically encountered in tissues [17,18,19]. Physiological oxygen for mammalian cells ranges between 3% (such as in brain tissue) [20] and up to 14% in the arterial arm of the pulmonary microvasculature [21,22,23,24]. While hypoxia can occur as a result of increased metabolic demands (e.g., exercise, regeneration, inflammation), hyperoxia is usually only encountered by elective use of oxygen for therapeutic purposes. These include intra- and post-operative hyperoxia, frequently employed to increase neutrophil-mediated bactericidal activity, presumably by increased ROS production [25,26]. Hyperbaric oxygen treatments are also used for refractory diabetic wounds and to potentiate radiation efficacy [27,28,29,30], as well as during activities such as diving [31]. Hyperoxia, therefore, represents a non-physiological stimulus, for which no evolutionary adaptation has evolved. Although clinically important, the effects of hyperoxia on MVEC function and plasticity have not been investigated, with the exception of studies on lung microvasculature [17,32,33,34]. However, due to its direct interaction with the external environment, the pulmonary vasculature is seemingly suited to respond to hyperoxygenation, which is unlikely to be the case in any other microvascular network.

MVEC responses are presumed to be unique to the tissue they reside in; therefore, in this study we used murine MVEC isolated from two continuous capillary networks from distinct microenvironments, lung and brain, and compared their responses to hypoxia following different culture conditions. We found that environmental oxygen priming significantly alters the MVEC response to hypoxia, compromising perception, responses, adaptation and survival due to compromised timing and amplitude of metabolic shifts; these alterations are organ-specific and appear to have more severe effects in MVEC originating from the brain.

## 2. Materials and Methods

Please refer to Appendix A for comprehensive list of reagents and equipment suppliers, source and catalogue numbers.

### 2.1. Isolation of Brain MVECs

Brain MVEC (*b*MVEC) were isolated by incorporating and slightly modifying previously described methods [35,36,37,38]. Brains of 6–8 wk old male C57BL6/J mice (4–6 animals per isolate), were excised and stored in serum-free DMEM (Sigma, St. Louis, MO, USA) on ice before surgical removal of the olfactory bulbs, cerebellum and mid-brain white matter. The remaining cortical tissue was rolled on sterile filter paper and subsequently digested in DMEM containing 2 mg/mL collagenase A (Roche, Basel, Switzerland) and 10 µg/mL DNase I (Roche) at 37 °C for 1 h, with gentle rotation. Digested tissue was pelleted at 290× *g* and resuspended in DMEM containing 20% BSA (Sigma) (*w/v*), then myelin fraction was separated by centrifugation at 1000× *g* for 10 min. The cell pellet was resuspended and filtered through a 70 µM nylon mesh and collected following centrifugation at 290× *g*. Cells were further digested in 2 mg/mL collagenase/dispase (Roche) and 10 µg/mL DNase I at 37 °C for 30 min. Following one wash in DMEM at 290× *g*, bMVEC were selected in medium supplemented with 4 µg/mL puromycin dihydrochloride (Sigma) [39,40] for the first 4 days.

### 2.2. Isolation of Lung MVEC

Lung MVEC (*l*MVEC) were isolated as described previously [41] with slight modifications. Lungs were excised from the same mice from which *b*MVEC were isolated. Lung tissue was digested for 90 min at 37 °C in HBSS (ThermoFisher Scientific, Waltham, MA, USA) containing collagenase A (2 mg/mL), supplemented with 2 mM CaCl_2_, 2 mM MgSO_4_ and 20 mM HEPES (Sigma). Cell suspension was filtered through a 70 µM nylon mesh and washed once in HBSS. Cell pellet was resuspended in PBS containing 0.1% BSA and incubated with anti-rat IgG Dynabeads (ThermoFisher Scientific) bound to rat anti-mouse CD31 antibody (BD Pharmingen^TM^, BD Biosciences, San Jose, CA, USA) for 90 min at 4 °C. The cells (bound to the beads) were washed three times with 0.1% BSA in PBS and plated in MVEC growth medium.

### 2.3. MVEC Culture and Hypoxia Treatment

All MVEC are maintained in collagen-coated TC plates in MVEC culture medium (1:1 mixture of low-glucose DMEM and F12 HAM nutrient mixture (Sigma), buffered with 20 mM HEPES and supplemented with 1% nonessential amino acids (Sigma), 2 mM sodium pyruvate (ThermoFisher Scientific), 20% FBS (Gibco, ThermoFisher Scientific, Waltham MA, USA), 75 µg/mL endothelial cell growth supplement (Sigma) and 100 µg/mL heparin (Sigma) and in atmospheres containing 5% CO_2_ and either ambient (~21%) or physiological O_2_—corresponding to in vivo levels in the lung (10%) and brain (5%) [20,42]. All media and solutions were pre-equilibrated with the appropriate oxygen concentration 12 h prior to media changes, trypsinization or cell treatments. All primary cells were used for experiments at no later than passage 3. All hypoxia experiments were carried out in an atmosphere containing 5% CO_2_ and 1% O_2_ at 37 °C, with controlled humidity, using either a Ruskinn Sci-Tive (Baker, UK) or a Whitley H35 HEPA hypoxystation (don whitley Scientific, Bingley, UK). Cells received fresh medium before hypoxia treatments, which was pre-equilibrated for 24 h at 1% O_2_ [43].

### 2.4. Viability Time Course Using Propidium Iodide

Brain and lung MVEC were expanded to 90% confluence before being transferred to 1% O_2_. At each timepoint, cells were washed once in PBS, detached using 0.25% trypsin, and resuspended in fresh media. All reagents used had been pre-equilibrated to 1% O_2_. Cell viability was assessed using an Adam^TM^-MC automated cell counter (NanoEnTek, Seoul, South Korea) according to manufacturer’s instructions. Cells were incubated with either a total cell stain, containing Propidium Iodide (PI) and a lysis agent, or with a non-viable cell stain, containing only PI. Viability was measured as a percentage of non-viable cells compared to total cell number.

### 2.5. Real Time Viability

Real time viability was assessed using the RealTime-Glo^TM^ MT Cell Viability Assay (Promega, Madison, WI, USA) according to manufacturer’s instructions. An amount of 3000 cells were plated one day before the start of the assay on collagen-coated white 96-well plates. Immediately before the start of the assay, MT Cell Viability Substrate and NanoLuc enzyme were added to growth medium pre-equilibrated to the appropriate O_2_. Luminescence was read every 10 min over 48 h, using a FLUOstar Omega plate reader set to 37 °C, 5% CO_2_ and at the appropriate oxygen level using an Atmospheric Control Unit (BMG Labtech, Ortengberg, Germany).

### 2.6. qPCR

Brain and lung MVEC at 90% confluence were either transferred into a hypoxia chamber containing 1% O_2_ or maintained at the same oxygen concentration. RNA was isolated using the RNeasy isolation kit (Qiagen, Hilden, Germany) according to manufacturer’s instructions. cDNA was synthesised from 1 µg of RNA using SuperScript III reverse transcriptase (Invitrogen, Waltham, MA, USA) according to the manufacturer’s instructions. All transcript levels were measured in triplicate and in a minimum of three biological replicates (cell batches); target transcripts were normalised to β-actin (Sigma); fold-change was calculated in relation to the 4 h baseline reading.

### 2.7. Western Blot Analysis

Brain and lung MVEC were expanded to 90% confluence. The cells were then either transferred into a hypoxia chamber containing 1% O_2_ or maintained at the same oxygen concentration in which they were cultured since isolation. In both cases, a media change was carried out at t = 0 using equilibrated growth medium. Cytoplasmic and nuclear protein were collected after the indicated times using NE-PER Nuclear and Cytoplasmic Extraction Reagents (ThermoFisher Scientific) according to manufacturer’s instructions. Total protein was extracted with RIPA buffer containing protease inhibitors (Roche). Protein concentration was measured using a Pierce™ BCA Protein Assay Kit (ThermoFisher Scientific). Unless specified otherwise, the amount of protein used per lane was 3 µg for nuclear protein and 20 µg for cytoplasmic and total protein. Samples were resolved in 3–8% Tris-Acetate gels or 4–12% Bis-Tris gels, and subsequently transferred to PVDF membranes using semi-dry blotting cassettes (Power Blotter XL, ThermoFisher Scientific or Trans-Blot Turbo, Bio-Rad, Hercules, CA, USA). The membranes were probed with primary antibodies o/n at 4 °C followed with HRP-conjugated secondary antibodies for 1 h at room temperature. All antibodies were diluted in PBST containing 2% milk. Target bands were detected with Pierce ECL Western blot substrate (ThermoFisher Scientific), and quantification was performed using ImageJ. Multiple loading controls were tested and compared for different sample types, and β-actin consistently showed equal loading across blots; this was seen also in nuclear extracts, due to an expected 10% cytoplasmic contamination (as per manufacturer’s protocol) and was the preferred loading control in these samples for its reliability (see Appendix A for the validation of β-actin as loading control for nuclear extracts).

### 2.8. Western Blot for Mitochondrial Electron Transfer Chain Complexes

Of protein, 35 µg (extracted and quantified as above) were resolved on 4–12% Bis-Tris gels and transferred to PVDF membranes. Protein samples were not heated prior to gel-separation to prevent damage to mitochondrial complex I, unstable above 50 °C. Total OXPHOS Rodent WB Antibody Cocktail (ThermoFisher Scientific) was diluted 1:250 and WB performed in PBST containing 1% milk. Bands were visualized using Amersham ECL Western Blotting Detection Reagent (GE Healthcare, Chicago, IL, USA). Rat heart mitochondrial extract was used as a positive control.

### 2.9. Glucose Uptake Assays

Of MVEC, 3 × 10^4^ were seeded per well in a 96-well plate 12 h before the start of the assay and either transferred into a hypoxia chamber containing 1% O_2_ or maintained at the same oxygen concentration. The start of each hypoxia incubation was staggered such that 2-DG treatment could be carried out at the same time for all conditions. The cells were washed once with PBS and incubated with 0.1 mM 2-DG in PBS for 10 min. Glucose uptake was measured using the Glucose Uptake-Glo^TM^ Assay (Promega, Madison, WI, USA) according to the manufacturer’s protocol. All reagents used were equilibrated to the appropriate O_2_ prior to the assay. Samples were incubated with detection reagent for 2 h and luminescence was measured with a FLUOstar Omega plate reader (BMG Labtech). A standard curve of 2-deoxy-glucose-6-phosphate was used to calculate glucose uptake rates. Data are normalised to cell number.

### 2.10. Glycolytic and Mitochondrial Stress Tests

Local pH and O_2_ changes in media were measured using a XFe96 Analyzer (Agilent Technologies, Santa Clara, CA, USA). Appropriate cell density was optimised for each measurement and condition to avoid anoxia after FCCP (Sigma) treatment during hypoxia treatments; 8 × 10^3^ MVEC were used for mitochondrial stress tests at baseline, and 5 × 10^3^ at 1% O_2_. For glycolytic stress tests, 10 × 10^3^ cells were used. MVEC were plated on Seahorse microplates precoated with collagen I and left to adhere for 12 h, then assayed immediately (baseline readings) or transferred to 1% O_2_ for 24 h before the assay. Immediately prior to transfer to hypoxia, all cells received fresh and pre-equilibrated medium. Before mitochondria stress tests, all cells were washed twice with Seahorse XF base medium supplemented with 7.8 mM glucose, 2.5 mM glutamine and 3.5 mM pyruvate. For glycolytic stress tests, Seahorse XF base medium was supplemented with glutamine only. The pH of the medium was adjusted to 7.4 using 1 M NaOH. The cells were then equilibrated with assay medium for 45 min at 37 °C in a CO_2_-free atmosphere at the appropriate O_2_ level. After recording baseline measurements, for mitochondrial stress tests, 1 μM Oligomycin (Sigma), 1 mM FCCP and combined Antimycin A (Sigma) and Rotenone (Sigma) (0.5 μM each) were added successively. For glycolytic stress tests, successive injections of 10 mM Glucose (Sigma), 1 μM Oligomycin and 50 mM 2-deoxy glucose were sequentially injected. Changes in pH and oxygen consumption were tracked in real time after each injection. Each measurement was composed of a mix–wait–measure cycle of 5–1–2 min (XF24 in hypoxia and normoxia), 5–0–2 min (XFe96 in hypoxia), or 3–0–3 min (XFe96 in normoxia), as per the manufacturer’s recommendation. Extracellular acidification rates (ECAR) and oxygen consumption rates (OCR) were normalised to the µg of total protein and quantified with BCA Protein Assay (ThermoFisher Scientific) after each assay.

### 2.11. Real Time Hypoxia Response Curves

Real time hypoxia response curves were measured using a Seahorse XF24-3 analyser. Amounts of 3 × 10^4^ MVEC per well were plated the day before the assay on Seahorse microplates precoated with collagen I. At the start of the assay, MVEC growth medium was replaced with Seahorse XF base medium supplemented with glucose (7.8 mM), pyruvate (3.5 mM) and glutamine (2.5 mM), matching the concentrations found in MVEC growth medium. Seahorse medium was equilibrated to 1% O_2_ for 24 h prior to the start of assay, and ECAR was measured every 20 min for 24 h. Each measurement performed at 5–1–2 min mix–wait–measure cycle, followed by a time delay of 12 min. As above, ECAR was normalised to the µg of total protein.

For experiments at non-atmospheric O_2_ conditions, the XF24-3 or XFe96 Analyzer was placed in a temperature-controlled, humidified gas flow-controlled chamber, in a CO_2_-free atmosphere and O_2_ levels were set to either 10%, 5% or 1%. All media and calibrant were equilibrated to the desired assay O_2_ atmosphere for 12 h prior to the assay. To avoid reoxygenation, cell culture plates were transported from their incubators to the instrument chamber in an air-tight container, and all washes were carried out within the chamber.

### 2.12. Mitochondrial Staining

MVECs were seeded on sterile glass chamber slides precoated with collagen I and grown to confluence. The cells were stained with fresh O_2_-equilibrated media containing 200 nM MitoTracker Red CMXRos (ThermoFisher) for 30 min, or DMSO as a negative control. The wells were washed twice with PBS and the cells were fixed with cold acetone (−20 °C) for 10 min and permeabilised with 0.5% Triton-X100 in PBS for 5 min at room temperature. Blocking was carried out using 10% donkey serum in PBS for 1 h at room temperature. Primary VE-Cadherin antibody was diluted in blocking solution (1:50, R&D Systems, Minneapolis, MN, USA) and the slides were probed over night at 4 °C, followed by a secondary antibody (Goat IgG conjugated with AlexaFluor 488) at room temperature for 1 h. Slides were mounted using ProLong Diamond Antifade with DAPI (ThermoFisher). All images were obtained with a Leica DM5500 B Fluorescence Microscope (Leica, Milton Keynes, UK) and analysed using ImageJ (National Institutes of Health, Bethesda, MD, USA).

### 2.13. Migration Assays

Fully confluent MVEC in a 12-well plate were treated for 2 h with 10 mM Mitomycin C (or equal volume of vehicle control) in adequately O_2_-equilibrated, serum-free growth medium. A scratch was made along the well using a P1000 tip and wells washed twice with PBS to remove detached cells; cells were then incubated with fresh, oxygen-equilibrated growth medium and imaged immediately at the start of the assay (0 h) and subsequently at 2, 4, 8, 24 and 48 h. The area of the scratch was measured using ImageJ and normalised to the area at T = 0 h to assess percent closure.

### 2.14. Succinate Dehydrogenase Assay

Extracts and assay were performed using a commercial succinate assay kit (Abcam, Cambridge, UK), with minor modifications. The confluent wells of 6-well plates were rinsed in ice-cold PBS and resuspended in 100 μL of ice-cold SDH Assay Buffer (provided), both pre-equilibrated to the adequate O_2_ levels (21%, 10%, 5% or 1% O_2_). Cells were scrapped from wells using cell lifters, then transferred to pre-cooled microfuge tubes and homogenized by pipetting. Clear extracts were used for assay, and the conversion of substrate (DCIP, provided) was assessed by following changes in absorbance at 600 nm over time. Measurements in kinetic mode were taken continuously for 30 min, and a linear range was selected to calculate the changes of absorbance over time (DOD/DT). DOD was converted to nmol of DCIP following the subtraction of background and against a standard curve. Results were normalized to amount of protein used per well, quantified by BCA.

### 2.15. Succinate Assay

Extracts and assay were performed using a commercial succinate assay kit (Abcam) with minor modifications. Confluent wells of 6-well plates were rinsed in ice-cold PBS and resuspended in 100 μL of ice-cold succinate assay buffer (provided), both pre-equilibrated to the adequate O_2_ levels (21%, 10%, 5% or 1%). Cells were scrapped from wells using cell lifters, then transferred to pre-cooled microfuge tubes and homogenized by pipetting. Cleared supernatant was deproteinated through a 10 kDa filter column. Filtrates (10 μL) were used for assay, and background wells (lacking succinate converter) were run in parallel for each sample. Plates were incubated for 30 min at 37 °C and the endpoint measurements taken at 450 nm. The quantification of succinate was performed against a standard curve, after background correction and subsequently normalized to the amount of protein used in each well (calculated from BCA assay, performed prior to deproteination).

### 2.16. Statistical Analysis

Data were analysed in Prism Graphpad 9 (Graphpad, San Diego, CA, USA) replicate number and statistical tests are described for each experiment.

Please refer to summary table of reagents for details of source and suppliers.

## 3. Results

### 3.1. Brain and Lung Microvascular Endothelial Cells Respond Differently to Hypoxic Stress

Primary murine brain and lung MVEC were cultured in standard atmospheric conditions (18–21% O_2_) and subsequently exposed to hypoxia (1% O_2_) for up to 48 h. Cell viability was shown to decrease in both MVEC over time (Figure 1a) but the effect occurred earlier and was more visibly pronounced in *b*MVEC. Similarly, a real-time viability assay [44] showed a more severely decreasing *b*MVEC growth curve (Figure 1b).

As known key mediators of hypoxia response, the activation of the two main HIF-α isoforms (HIF-1α and HIF-2α) was assessed in both MVEC populations to investigate if discrepancies in the HIF-signalling pathway corresponded to differences in cell viability. HIF-1α expression showed the canonical transient upregulation in both lung and brain MVEC [45,46,47], although consistently higher levels of this isoform were found in cells from the lung at all time points (Figure 1c). HIF-2α protein levels decreased only at later time points for *l*MVEC, but *b*MVEC showed strikingly higher levels of HIF-2α protein in all conditions (Figure 1d).

Selected transcriptional HIF targets were quantified by RT-qPCR, and glycolytic transripts *PGK*, *LDH-A* and *GLUT1* were upregulated in both MVEC, suggesting that HIF-α isoform preference does not necessarily affect or condition the transcriptional activation of typical hypoxia-induced genes.

Even though the relative changes differed for each target, such as higher induction of *PGK* mRNA in *l*MVEC or the earlier upregulation of *LDH-A* in *b*MVEC, after 24 h of hypoxia the fold induction was no longer different between cell populations. One interesting distinction is that hypoxic *l*MVEC accumulated *VEGF* and *ARG2* mRNA in all hypoxia conditions, but levels of either of those transcripts were seen to decrease in *b*MVEC (Figure 1e). This was particularly unexpected for *ARG2*, which is indeed completely undetected in *b*MVEC after 48h of hypoxia, if the cells had been previously primed at room oxygen levels (Figure 1c) even though this this transcript is primarily regulated by HIF-2α [48], the isoform that is preferentially present in this cell population. *BNIP3*, a HIF-1α target [49,50,51], was upregulated more strongly in brain than *l*MVEC, which correlates with the higher susceptibility to hypoxia-induced cell death seen in cells isolated from the brain (Figure 1a,b) but not with HIF isoform preference.

These data show that MVEC from the brain and lung display markedly distinct responses to hypoxia, including survival and HIF isoform stabilization, but the differences in hypoxia-driven transcriptional activation was not as strongly affected. In vivo average oxygen tension in brain tissue (~5% O_2_) [20] is much lower than that found in the lung (~10%) [42], implying that *b*MVEC would be better equipped to survive an exposure to 1% O_2_. Paradoxically, *b*MVEC were more adversely affected by hypoxia, indicated by their reduced viability, and these cells appeared unable to mount a suitable adaptive response when transferred to hypoxia, even though for many tissues, including the brain, 1% O_2_ can still be considered physiological [52,53].

Crucially, however, both organs contain much less oxygen than that found in normobaric room air [54], and thus the cellular responses observed at atmospheric oxygen tensions (~21% O_2_) are unlikely to represent physiological adaptations to hypoxia in tissue.

Hence, the experiment was redesigned to assess whether the cells’ response was affected by previous hyperoxia stress. To investigate this, primary MVEC were subsequently expanded in what was determined to be the best approximation to their typical physioxic state.

### 3.2. Oxygen Priming Differentially Conditions the Hypoxia Response of Brain and Lung MVEC

MVEC were cultured at either 10% or 5% O_2_ (physiological for lung and brain, respectively [23]) in addition to standard 21% O_2_, and response to hypoxia (1% O_2_) was evaluated for the same parameters as above. Here, we compared viability, hypoxia response and transcriptional regulation within different oxygen-priming conditions as well as across the two cell populations. Upon transfer to 1% O_2_, the viability of both MVEC populations cultured in physioxia remained higher than those primed at 21% O_2_ (Figure 2a,b). HIF-a isoform levels were again assessed after 4 h of hypoxia. As before, HIF-1α signal was consistently higher in lMVEC than bMVEC, and this was significant when cells were expanded at 21% and 10% O_2_ (but not 5%) prior to hypoxia. In both cell populations, HIF-1α levels after hypoxia were highest in MVEC cultured at 10% O_2_ irrespective of their tissue of origin (Figure 2c). The quantification of HIF-1α is shown in Figure 2e, where each data point refers to one biological replicate and dashed lines link lung and brain MVEC from the same experiment (and quantified from the same gel). Conversely, and similarly to what was shown in Figure 1d, HIF-2α protein levels were consistently higher in hypoxic bMVEC compared to lMVEC (Figure 2d). The quantification of the hypoxic HIF-2α signal is plotted in Figure 2f and, as above, signals from same experiment are linked by a dashed line. Significant differences between MVEC populations were only seen at the two highest O_2_ concentrations.

To investigate if these results reflected a change from the baseline HIF-α levels or were consistently high or low in specific MVEC populations, HIF protein levels before hypoxia were also quantified (Appendix A). As expected, both isoforms were stable at physiological oxygen, underscoring their role in normal EC function [24]; *l*MVEC has higher levels of HIF-1α, whereas HIF-2α at baseline was low in both MVEC. Additionally, and to allow a direct assessment of HIF-α change of signal following hypoxia, hypoxic and physioxic nuclear protein extracts were run on the same gel (Appendix A), confirming that the hypoxia induction of HIF-1α occurs primarily in *l*MVEC, except when they are expanded at 5% O_2_. This suggests that 5% O_2_ is sub-physiological (or even hypoxic) for *l*MVEC.

Contrary to the association of HIF-1α activation with the early hypoxia response, in *b*MVEC, the 4 h of hypoxia result in a higher induction of HIF-2α, even when compared to the extent to which *l*MVEC induce HIF-1α. The modest fold induction of HIF-1α in *l*MVEC exposed to hypoxia is a result of elevated HIF-1α levels in physioxic atmospheres, further demonstrating that the exposure of cells to high oxygen levels artificially removes a key transcriptional and metabolic regulator (HIF-1α), bound to affect the intrinsic biology of these cells and their subsequent responses. HIF-1α induction in *b*MVEC and HIF-2α induction in *l*MVEC seem independent of baseline O_2_ (Appendix A).

These data underscore the role of HIF transcription factors beyond that associated with oxygen deprivation and reinforce their relevance in homeostasis and metabolic plasticity in a myriad of physiological conditions [55,56,57]

The transcript levels of HIF targets are more strongly upregulated in cells maintained in physiological O_2_ (Figure 2g and Appendix A). Most strikingly, *VEGF* and *ARG2* mRNA levels were only induced by hypoxia in *b*MVEC if the cells had been cultured at 10% or 5% O_2,_ andARG2 completely undetectable in hyperoxia-primed *b*MVEC. The autophagy marker BNIP3, however, was induced most strongly in *b*MVEC cultured at 21% O_2_, correlating with their reduced survival to hypoxia in this condition. To extricate whether the mRNA fold-change was skewed by transcript levels at baseline, the relative abundance of hypoxia-responsive transcripts before hypoxia treatment was plotted (Appendix A). Indeed, some intrinsic differences are seen in the relative abundance of individual transcripts between the two MVEC populations, such as lower levels of *LDH-A* and *VEGF* in *b*MVEC. Baseline transcript levels were also different depending on O_2_ priming, and *l*MVEC maintained at 10% O_2_ had much higher *GLUT1* levels. Overall, the largest increase in transcript levels of hypoxia response genes were generally seen in cells grown in physiological oxygen, especially at 10% O_2,_ and mostly for *b*MVEC.

To investigate if discrepancies were due to the upstream regulation of HIF stabilisation (prolyl hydroxylases PHD1-3) or transcriptional activity (FIH) [58,59], relevant protein levels were investigated by Western blot (Appendix A). The protein levels of the canonical regulators of HIF were quantified, but no O_2_-dependent patterns or tissue-specific differences were seen to significantly differ between MVEC (Appendix A). PHD2 levels were slightly reduced in *b*MVEC at 10% and 5% O_2_, and PHD3 was consistently higher in those cells, in all O_2_ conditions, compared to *l*MVEC. FIH expression was significantly elevated in MVEC maintained in 21% O_2_, irrespective of the tissue of origin. While these results may not explain the HIF isoform protein levels, they may underlie the dampened hypoxia response seen in hyperoxic MVEC. The levels of *HIF-*α mRNA were also measured (Appendix A), and almost entirely reflect the baseline protein abundance quantified by WB, further confirming that HIF regulation does not occur exclusively post-translationally, as originally proposed [60,61,62]. However, *l*MVEC grown at 21% O_2_ displayed much lower levels of *HIF-1*α mRNA than any other sample, a trend not reflected at the protein level.

The data above show that MVEC viability is affected by oxygen priming, yet while HIF-α isoform activation is tissue-specific, this does not appear to affect the HIF-dependent transcriptional activation of hypoxia-related targets or viability when cells are primed in physioxia; as such, the different tolerance seen between lung and brain MVEC appear largely independent from HIF stabilization.

### 3.3. Glycolytic Activity of MVEC Is Altered as a Result of Oxygen Priming

Endothelial metabolism underlies endothelial function, and EC are widely believed to be intrinsically and primarily glycolytic [63]. Nonetheless, a further shift towards glycolysis is key to enable adaptation and survival to hypoxia. Glycolytic stress tests were performed to compare glycolytic metabolism in MVEC from brain and lung tissues, using extracellular acidification rate (ECAR) as a readout of glycolysis at baseline and upon complete mitochondrial inhibition with Oligomycin (maximal capacity) [64]. Representative graphs are shown in Figure 3a, and the summaries of average glycolytic parameters quantified in the two MVEC populations are presented in Figure 3b. Baseline glycolysis, as expected, was higher in cells maintained in environments with lower O_2_. Although maximal glycolytic capacity (and thus spare capacity) was comparable between the two MVEC at all O_2_ tensions, *l*MVEC had slightly but significantly higher basal glycolytic activity in phyxsioxia (5 and 10% O_2_) than *b*MVEC (Figure 3b).

Changes in glycolytic function after hypoxia stress were subsequently assessed (Figure 3c,d). Here, cells expanded in the three different oxygen environments were transferred to 1% O_2_ for 24 h prior to the assay to stabilise any metabolic changes resulting from adaptation to hypoxia. Cell viability was confirmed and cell density optimised for each MVEC population (see methods), to avoid assaying non-viable cells or induce anoxia in assay wells. Results show that glycolytic activity after adaptation to hypoxia is both oxygen-dependent and organ-specific (Figure 3c). *l*MVEC maintained an inverse correlation between O_2_ levels and glycolytic rates seen at baseline (Figure 3b). *l*MVEC showed a basal glycolytic rate lower than that of *b*MVEC if the cells were grown at 21%, similar in cells grown at 10%, and higher when the cells were grown at 5% O_2_ (Figure 3d). Interestingly, hypoxic *b*MVEC had a larger glycolytic spare capacity following 24 h of hypoxia in all conditions (significant at higher O_2_ levels). Indeed, *b*MVEC are much more likely to encounter such oxygen levels in vivo and thus, by maintaining glycolytic plasticity, would be better equipped to adjust to low O_2_. This, however, is in stark contrast with the fact *b*MVEC are more susceptible to hypoxia, albeit only when primed at 21% O_2_ (Figure 1 and Figure 2).

To further investigate this conundrum, ECAR was measured immediately upon transfer to 1% O_2_, such that the glycolytic shift could be observed in real-time, instead of allowing the cells 24 h to adjust. This revealed that oxygen priming drastically affects the timing and amplitude of MVEC adaptation to hypoxia (Figure 4a, note y-axis matched scale). Both lung and brain MVEC grown in 21% O_2_ (left) show a very minor increase in ECAR, which was further delayed in *b*MVEC. If primed at 10% O_2_ (middle), *l*MVEC increased glycolytic activity within the first 5 h to more than 2-fold compared with the same cell population expanded at 21 % O_2_. For the same conditions (21% and 10%), the increase in ECAR for *b*MVEC upon hypoxia exposure was very subtle. However, when MVEC were maintained at 5% O_2_ prior to hypoxia (right), the ECAR measured in *b*MVEC increased within minutes and to an extraordinary extent, the highest seen for either cell type in any condition. This experiment demonstrates that *b*MVEC are significantly more responsive to hypoxia but only if kept in physioxia. Conversely, *l*MVEC expanded in 5% prior to hypoxia stress were unable to further increase ECAR, supporting that 5% O_2_ is already perceived as hypoxia in the lung.

Irrespective of O_2_ availability, real-time measurement of glucose-uptake rates confirms those are intrinsically higher in *b*MVEC (Figure 4b and Appendix A), and indeed higher than suggested by *GLUT1* transcript levels (Figure 1e, Figure 2f and Appendix A). As expected, glucose uptake increases in all MVEC upon transfer to hypoxia (Figure 4b), except in *b*MVEC primed at 21% O_2_ (Figure 4b, left), which appear inept at increasing glucose uptake until 8 h after hypoxia exposure. Glucose uptake is an essential step in triggering a concomitant increase in glycolytic activity and is in agreement with the data Figure 4a (left), showing that *b*MVEC have a delayed perception of O_2_ levels when maintained in 21% O_2_. This result shows that, while *b*MVEC preserve glycolytic capacity in all atmospheres, when maintained in hyperoxia they fail to source the substrate essential to actually performing glycolysis within a relevant time frame, which likely compromises survival.

To investigate whether the increased hypoxic viability of MVEC grown at lower oxygen levels was indeed due to dependence on glycolytic metabolism, viability assays were performed in the presence of 10 mM 2-DG, a competitive inhibitor of glycolysis. Removing the ability to use glucose for cellular energetic demands under hypoxia resulted in the arrest of proliferation, irrespective of the glycolytic reserve of each MVEC population or culture condition (Figure 4c).

Combined, these results strongly indicate that supra-physiological oxygen conditions impair MVEC perception of, and therefore the response to, a hypoxic challenge. It further shows that this phenomenon is more pronounced in MVEC that originate from brain tissue.

### 3.4. Effects of Oxygen Priming on Mitochondrial Respiration

Following the assessment of glycolytic capacity and response in different MVEC populations, the effect on mitochondrial metabolism was investigated and compared as a function of oxygen availability. Oxygen consumption rates (OCR) were calculated from mitochondrial stress tests. Mitochondrial basal and maximal respiration rates, predictably, directly correlated with oxygen levels in both MVEC: those grown at 21% O_2_ have the highest OCR, whereas those maintained at 5% O_2_ were the least reliant on mitochondrial respiration, as their metabolic preference is, naturally, glycolysis (Figure 5a, note mismatched y-axis scales, and Figure 5b). This pattern was considerably more pronounced in *l*MVEC, and as a result the maximal OCR of *l*MVEC grown at 21% O_2_ was higher than that of *b*MVEC (Figure 5a,b, left); at 5% O_2_ that pattern was strikingly reversed (Figure 5a and b, right), and *b*MVEC instead retained a proportionally higher mitochondrial reserve. At 10% O_2_ (Figure 5a,b, middle panels) there was no difference between the two MVEC populations. Thus, at physiological O_2_, *b*MVEC maintain higher relative mitochondrial spare capacity than their lung counterparts.

Mitochondrial respiration parameters were assessed again but following adaptation to hypoxia (after 24 h at 1% O_2_). Similarly to what was seen at baseline (prior to hypoxia), basal respiration rates were identical between the two MVEC populations primed at 21% O_2_ (Figure 5c,d, left). Maximal respiration was seen to decrease only in *l*MVEC, whereas bMVEC grown in physiological O_2_ retained a significantly higher maximal respiration rate compared to lMVEC (Figure 5c,d, middle and right panels).

These data indicate that *b*MVEC have wider metabolic plasticity, and overall higher mitochondrial activity than their lung counterparts. However, it is also evident that *b*MVEC are more vulnerable to high oxygen levels.

To confirm that the EC metabolic profile is reflected in functional aspects of MVEC behaviour, a migration assay was performed to assess alterations in EC function as a result of O_2_ priming (Appendix A). This was carried out in serum-free medium (Appendix A) or also in the presence of the proliferation inhibitor Mitomycin C (MM) [65] (Appendix A), to discriminate closure due to proliferation and migration, or migration alone. MVEC maintained in their respective physioxia closed the scratch wound more effectively than in either of the alternative atmospheres and were slowest to migrate when in hyperoxia. *b*MVEC were generally less motile than those from lung, and the effect of O_2_ pre-conditioning was less pronounced; additionally, seeing that the fastest closure occurred in cells at 5% O_2_ suggests that *b*MVEC are intrinsically less migratory and rely more on cell division for wound closure than *l*MVEC do and appear generally less susceptible to the presence of MM.

### 3.5. Effects of O_2_ on Mitochondrial ETC Complexes Is Organ-Specific

Having investigated the effects of varying oxygen levels on mitochondrial metabolism, the impact on the mitochondrial ETC composition was further examined. Total protein was isolated from *b*MVEC and *l*MVEC at baseline and following 24 h of adaptation to 1% O_2_, for each oxygen condition. These were resolved by SDS-PAGE, probed with a mitochondrial ETC antibody cocktail [66] and quantified upon normalization against a positive control (Figure 6). Representative blots of *l*MVEC and *b*MVEC mitochondrial protein are shown in Figure 6a and b, respectively, and the ratio of signal for each complex was compared to that seen at 21% O_2_ (Figure 6c,d). In *l*MVEC, there was a positive (and expected) correlation between oxygen levels and mitochondrial complex protein levels, i.e., cells expanded at lower O_2_ also had, correspondingly, lower levels of mitochondrial ETC complexes than those maintained at 21% O_2_. This was further (and in most cases, significantly) exacerbated in *l*MVEC following adaptation to 1% O_2_ for 24 h (shaded areas), mostly in cells expanded in physioxia.

A very different pattern was seen in MVEC that originated from brain tissue (Figure 6b,d); instead of a general decrease in protein from all ETC complexes, *b*MVEC cultured at 10% or 5% O_2_ have much higher levels of Complex II, which increased following hypoxia exposure (Figure 6d). Notably, if *b*MVEC were kept in hyperoxia (21% O_2_), complex II levels were not changed by hypoxia (Figure 6d). The activity of succinate dehydrogenase (SDH, complex II) (Figure 6e,f) as well as succinate levels (Figure 6g,h) were quantified at baseline and following adaptation to hypoxia, in MVEC primed in their own physiological oxygen and compared to those maintained in normal atmosphere—note y-axis scales range. In *b*MVEC, SDH activity correlated with SDH protein levels shown in Figure 6b,d and show a striking and inverse correlation with succinate levels, which accumulate to significantly higher concentrations in cells from 21% O_2_, both at baseline and after hypoxia. In physioxic (but not hyperoxic) *l*MVEC, SDH activity is not affected by culture conditions or hypoxia, but succinate levels (Figure 6g) increase following hypoxia. Interestingly, both baseline levels and the extent of accumulation of succinate in these cells are nearly an order of magnitude lower than what is seen in *bMVEC*. These data suggest that mitochondria from hyperoxic *b*MVEC lose SDH protein and activity, and as a result accumulate high and potentially toxic levels of succinate. Basal SDH activity and changes in succinate levels in *l*MVEC change within a much lower scale, indicating that this is a finely tuned but likely less vital aspect of mitochondrial function in MVEC originally from the lung.

The signal intensity for all other quantifiable complexes was not different between any conditions in *b*MVEC, with the notable exception of much lower levels of complexes III and I in hypoxic cells that were primed at 21% O_2_. These two complexes are in fact the two mostly associated with the ROS generation downstream of mitochondrial activity.

The quantification of mitochondrial potential using MitoTracker Red CMXRos, in which higher signal intensity correlates with mitochondrial health and activity (Appendix A), indeed suggested that *b*MVEC have lower mitochondrial activity at 21% O_2_, and qualitative observation indicates that hyperoxic mitochondria in lMVEC are more active than those from *b*MVEC. This could be a result of oxidative damage, to which *b*MVEC mitochondria appear considerably more susceptible.

These opposing oxygen-induced changes in ETC composition (Figure 6) and mitochondrial activity (Appendix A) between the two MVEC populations are consistent with the measurements of maximal mitochondrial respiration shown above (Figure 5), where in *b*MVEC, higher maximal OCR occurred at lower baseline oxygen levels (Figure 5a,b, right) and was higher than that of lMVEC after adaptation to hypoxia in cells previously maintained in physiological oxygen. Conversely, when *l*MVEC were maintained in physiological O_2_, they lost their mitochondrial spare capacity almost entirely (Figure 5c,d, middle and right panels).

## 4. Discussion

MVEC are key regulators of organ homeostasis and locally control perfusion and permeability to match specific tissue requirements [1,3]. Perception and response to oscillations in oxygen availability, whether as a result of changes in tissue need or environmental supply, are essential aspects of MVEC function [46]. In this study, primary MVEC from two continuous endothelial capillary networks were isolated from brain and lung tissue and their responses to hypoxia compared.

The distinct responses seen here demonstrate that EC are not only intrinsically heterogeneous [5,49,50] but functionally and metabolically reprogrammable by exposure to different O_2_ conditions.

The canonical transcriptional hypoxia response is known to rely on the activation of the HIF pathway and the stabilization of either or both main α isoforms. Brain and *l*MVECs differentially express HIF-α isoforms, with *l*MVECs consistently stabilizing higher levels of HIF-1α both at baseline physioxia and after hypoxia; while HIF-1α is usually associated with acute hypoxia response [67,68] the preferred isoform stabilized in *b*MVEC is HIF-2α, induced by hypoxia to a much higher degree than that seen in *l*MVEC. Regardless of the preferred HIF-α isoform stabilised, the mRNA levels of HIF transcriptional targets, specifically those encoding typical hypoxia response genes (e.g., *PGK*, *GLUT1*, *LDH-A* and *VEGF*), were induced to comparable levels in both MVEC populations (Figure 2), indicating that the distinct vulnerability to hypoxic insult in the two cell populations is largely HIF-independent. The factors regulating MVEC metabolic state and adaptation function of oxygen availability therefore remains to be elucidated and will be scrutinized in subsequent studies.

Endothelial function is intrinsically linked to endothelial metabolism [68,69,70]. In this study, baseline metabolic preferences and hypoxia-driven metabolic shifts were measured in the two MVEC and compared to investigate whether their distinct responses and tolerance resulted from intrinsic heterogeneity or oxygen priming.

As expected, baseline glycolytic activity was higher MVEC expanded in lower O_2_. Both MVEC grown in physiological atmospheres (5 or 10% O_2_) had increased maximal glycolytic rates (Figure 3), which correlated with increased viability during hypoxia (Figure 2). As hyperoxygenation impaired the ability to upregulate glycolysis during a subsequent hypoxic stimulus, even in the absence of mitochondrial respiration (Figure 3), the differences in glycolytic capacity may underlie the loss in cell viability. The timing and magnitude of the metabolic shift upon hypoxia stress is not only highly dependent on O_2_ priming but inherent to the organ of origin (Figure 4). *b*MVEC expanded in 5% O_2_ are strikingly more responsive when transferred to 1% O_2_ than if maintained at 10% or 21% O_2_ prior to hypoxia, despite a comparatively milder challenge (from 5 to 1% O_2_) and, in fact, the shift to glycolysis is severely compromised in *b*MVEC cells cultured at 21% O_2_ (Figure 5).

Interestingly, baseline glucose uptake is not affected by environmental O_2_ and is consistently higher in MVEC from the brain, reflecting the role of this microvascular network in the active shuttling of glucose to neural tissue [71,72,73]; however, the hypoxia-induced increase in glucose uptake in *b*MVEC is dramatically delayed in the cohort expanded in supra-physiological O_2_ levels, consistent with what was previously shown as the effect of hyperbaric oxygen on the brain vasculature [74], mirroring a diving experience. Combined, these results suggest that the hyperoxic priming of *b*MVEC has a more striking effect on their perception of a hypoxic challenge than on their capacity to respond to it.

While all EC are assumed to be primarily glycolytic, *b*MVEC are known to have significantly higher mitochondrial density [75], and thus presumably have higher mitochondrial activity. Remarkably, *b*MVEC only show significantly higher relative mitochondrial respiration rates than *l*MVEC when expanded in their own physiological atmospheres (5% O_2_). The dramatic lower maximal respiration of *b*MVEC relative to *l*MVEC, when expanded at 21% O_2_, shows that hyperoxia is much more damaging to *b*MVEC. These are also the O_2_ priming conditions at which *b*MVEC show a severe delay in the induction of glycolysis following hypoxia (Figure 5) and subsequent compromised viability (Figure 1 and Figure 2).

Mitochondria are evident sensors of O_2_ [76,77,78], and even though their cellular function is commonly associated with efficient generation of cellular ATP, in that process they become main generators of reactive oxygen species (ROS) [79,80,81]; these have been shown to mediate mitochondrial damage and associated high rates of cell death in bovine aortic EC exposed to high O_2_ [82]. Importantly, mitochondrial function has also been associated with the integrity of the BBB [83,84,85].

Mitochondrial ROS are generated primarily at complexes I and III of the mitochondrial electron transport chain (mETC) [78,81]. While in *l*MVEC all mETC protein complexes decrease almost linearly with decreased O_2_ availability, the only two complexes seen to decrease in hypoxic *b*MVEC are precisely complexes I and III, and only if the cells are primed in hyperoxia (Figure 6). Conversely, levels of complex II are higher in physioxic *b*MVEC, and increase following adaptation to hypoxia, again only if cells are never hyperoxygenated. This suggests that complex II activity is essential for the response in *b*MVEC to hypoxia, possibly in regulating succinate levels, which are detected at levels much higher than in *l*MEVC. Both SDH activity and succinate levels are dramatically affected by hyperoxia in *b*MVEC. The consequences of high succinate in these cells is not clear, but high succinate has been associated with the epigenetic regulation of gene expression, the molecular mediation of signal transduction and metabolic reprogramming [86,87,88]. Complex II is a unique transporter in the mETC, the only one exclusively encoded in the nuclear genome and which does not contribute to the proton gradient across the inner mitochondrial membrane [89]. Its role in EC has not been studied, and there is some ambiguity in the existing literature regarding the contribution of this complex to ROS formation [76,78,79,81,88]. However, such studies were performed in cells cultured in normal atmospheric O_2_, which is shown herein as hyperoxic for this cell population and where such changes would not have been observed (Figure 6d). To the best of our knowledge, this is the first instance where such assessments were performed while maintaining cells in physiologically relevant oxygen tensions. The underlying significance of high succinate levels and high activity of mitochondrial complex II in *b*MVEC is, however, unclear and worth further investigation, but our data indicate that the supra-physiological levels of O_2_ affect the stability of complex II in *b*MVEC, presumably corrupting reserve respiratory capacity (Figure 5) [90] and impairing ROS removal. This is supported by the fact that hyperoxygenated *b*MVEC become unresponsive to hypoxia.

If on one hand, having more mitochondria could make cells more sensitive to O_2_ oscillations (Figure 4, right panel), on the other hand, cells with more mitochondria will generate more ROS in direct correlation with O_2_ availability. Thus, the subsequent damage to the mitochondria will in turn compromise the ability to perceive O_2_ oscillations. This too is supported by the fact that *b*MVECs cultured at 21% O_2_ fare far worse than those from lung; not only do *b*MVEC undergo a more severe challenge relative to their physioxic setpoint, but brain EC also have more mitochondria [75] and higher baseline mitochondrial activity (Figure 5 and S6). While a similar effect can be seen in *l*MVEC, it is much less severe.

As *l*MVEC are exposed to wider oscillations of O_2_ due to their close proximity to air, they should be inherently more tolerant to higher O_2_ levels and oxidative stress. Most studies regarding the effects of oxygen in microvascular cells, specifically hyperoxia, are restricted to the lung [17,32,34,91] and occasionally arterial EC [92,93], where, according to our findings, microvascular networks from less oxygenated tissues should are likely to be much more susceptible.

It is noteworthy that hyperbaric oxygen treatments (HOT) used to treat refractory diabetic lower extremity wounds or radiation injuries result in an increase in O_2_ partial pressure of up to 15 times higher than standard air [28,29]; in rat brain tissue, this value translates to a 13-fold increase in oxygenation [27]. Visible acute side effects (seizures) affect very few patients [31], but early studies in rodents reported a 50% decrease in lung endothelium viability following exposure to pure O_2_ [32]. Even though the effects of hyperoxia on EC viability are less pronounced in primates, functional aspects have not been investigated [77]. With the unfolding of severe acute coronavirus syndrome 2 (SARS-CoV-2) in 2019/2020 pandemic, the use of supplemental oxygen has been also used to combat the resulting hypoxemia. However, there is no consensus on what the target blood oxygen saturation should be, with suggestions ranging from 90% up to >96% [94,95,96], with most treatment guidelines opting for intermediate values. Although the discussion hinges on patient mortality as the main indicator, if no differences in mortality are observed, our data suggest that the liberal use of oxygen should be met with caution. Previous data showed that perioperative hyperoxia increases mortality in patients with pre-existing pathologies [92,93,97], and so, provided our results translate into reprogramming of human microvasculature, the benefits of this practice should also be reassessed [98,99].

## 5. Conclusions

MVEC are highly responsive cells, with morphological and functional aspects that uniquely match the tissues they reside in; oxygen-sensing by these cells differs in sensitivity, timing and amplitude in different organs. While glycolytic capacity is essentially constant across organ sites, mitochondrial function is dependent on the tissue of origin.

Understanding the unique nature and response of tissue-specific microvasculature will allow the identification of particular organ vulnerabilities to oxygen oscillations resulting from physiological, pathological or therapeutic settings. Most molecular studies are performed in cells maintained in atmospheric oxygen, which this study unambiguously demonstrates is hyperoxic, which in turn affects baseline cell metabolism and subsequent cell behaviour.

Our data show that adaptation to hypoxia is compromised when cells are exposed to supra-physiological levels of oxygen: while glycolytic capacity is not affected, mitochondrial damage impacts the cell’s ability to perceive hypoxia in a timely fashion. In brain MVEC, this results in the loss of viability.

It further highlights the detrimental effects of hyperoxia, which are far less considered, albeit demonstrably no less harmful, underscoring the importance of research into organ-specific MVEC reprogramming by environmental challenges and the downstream effects in disease, treatment and long-term recovery.

The unique metabolism and function of organ-specific MVEC is shown to be reprogrammed by external oxygen. These findings establish the inadequacy of molecular and physiological studies performed at atmospheric oxygen tensions, while raising the need for refinement and functional outputs to confirm and measure the benefits of oxygen therapy.

## Figures and Tables

**Figure 1 cells-11-02469-f001:**
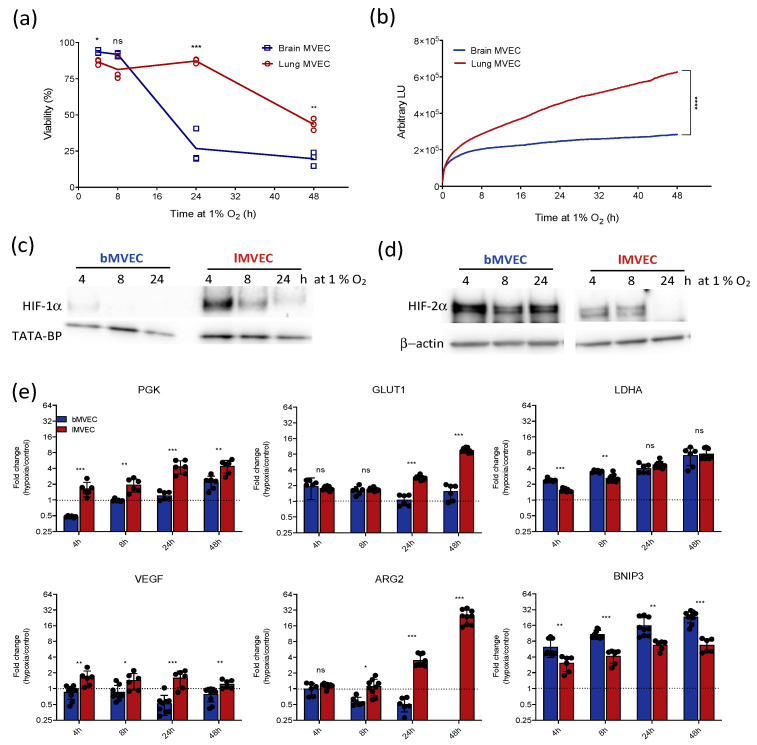
Lung and brain MVEC show organ-specific response and tolerance to hypoxia. (**a**) Viability of MVECs exposed to 1% O_2_ at t = 0, measured by propidium iodide staining (*n* = 3); t-tests corrected for multiple comparisons (Holm–Sidak) * *p* < 0.05, ** *p* < 0.01, *** *p* < 0.005), ns = not significant; (**b**) Real-time viability of brain and lung MVECs exposed to 1% O_2_ at t = 0, measured using the Real-Time Glo Assay (Promega) (*n* = 3). Shaded areas show SD, statistical differences were assessed by unpaired student’s t-test, using the area under the curve (**** *p* < 0.0001); (**c**,**d**) Western blot of HIF-1α and HIF-2α using nuclear extracts from brain and lung MVECs exposed to 1% O_2_ for the indicated amount of time; loading control TATA-BP to show nuclear extracts used; b-actin used subsequently because more consistent across samples and replicates (see Appendix A); (**e**) RT-qPCR for hypoxia targets *PGK*, *VEGF*, *GLUT1*, *BNIP3*, *LDH-A* and HIF-2α target *ARG2* in brain and lung MVEC, at 21%, 10% or 5% O_2_ baseline and after 1% O_2_. Average fold-change ± SD and individual readouts are shown (hypoxia/normoxia) (*n* ≥ 3). t-tests corrected for multiple comparisons (Holm–Sidak), * *p* < 0.05, ** *p* < 0.01, *** *p* < 0.001; ns = not significant.

**Figure 2 cells-11-02469-f002:**
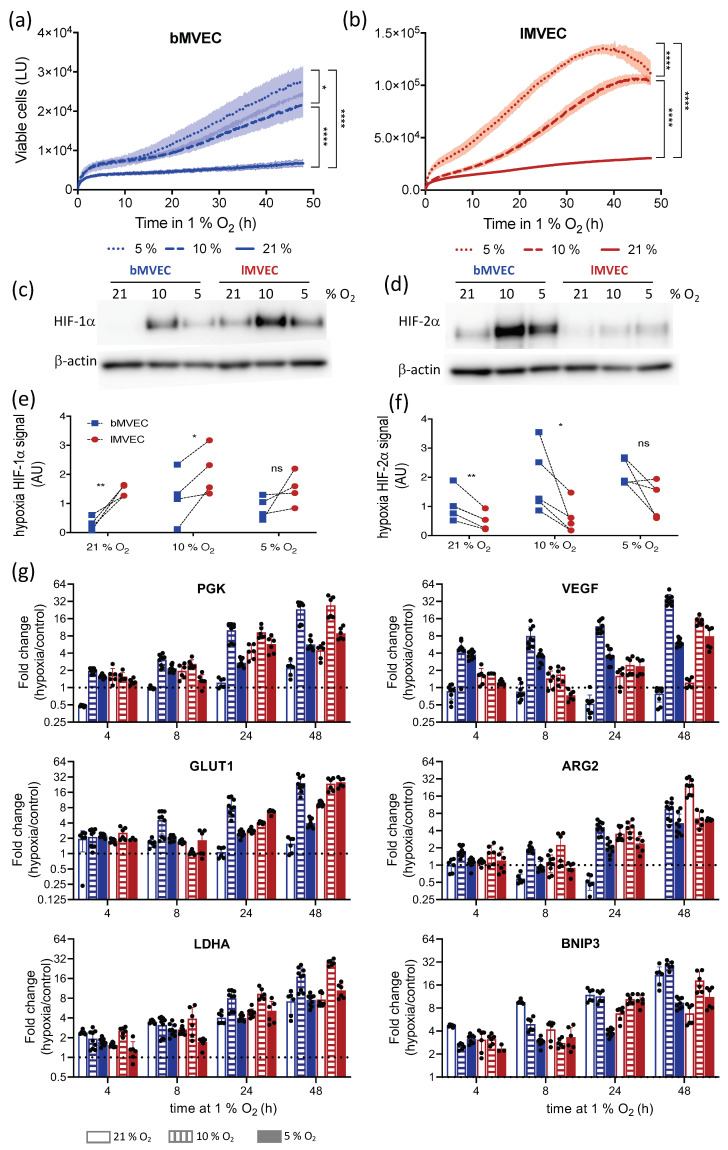
MVEC response to hypoxia is organ-specific and dependent on O_2_ priming. Real-time viability of (**a**) brain (blue) and (**b**) lung (red) MVEC maintained at 21%, 10% or 5% O_2_ and transferred to hypoxia (1% O_2_) at t = 0 was measured using a Real-Time Glo assay (Promega) (*n* = 3). Shaded areas show SD, and statistical differences were assessed by 2-way ANOVA with Holm–Sidak’s multiple comparison test, using the area under the curve (* *p* < 0.05, **** *p* < 0.0001, ns = not significant); (**c**,**d**) representative Western blot of HIF-1α using nuclear extracts from brain and lung MVEC, respectively, exposed to 1% O_2_ for 4 h, and previously expanded at 21%, 10%, or 5% O_2_; quantification of HIF-1α signal obtained by densitometry and normalized to the loading control is shown for brain (**e**) and lung (**f**); the lines connect samples from the same experiment (*n* = 4; paired t-test, * *p* < 0.05, ** *p* < 0.01, ns = not significant); (**g**) RT-qPCR for hypoxia-inducible transcripts *PGK*, *VEGF*, *GLUT1*, *BNIP3*, *LDH-A* and HIF-2α target *ARG2* in brain (blue) and lung (red) MVEC, primed at 21% (open), 10% (striped) or 5% (filled) O_2_ and exposed to hypoxia (1% O_2_) for different amounts of time. Data are shown as average fold-change (hypoxia/baseline) ± SD (*n* ≥ 3). Statistical differences were assessed by 2-way ANOVA with Holm–Sidak’s multiple comparison test, and comparisons between the organ of origin and across time points are summarized in Appendix A.

**Figure 3 cells-11-02469-f003:**
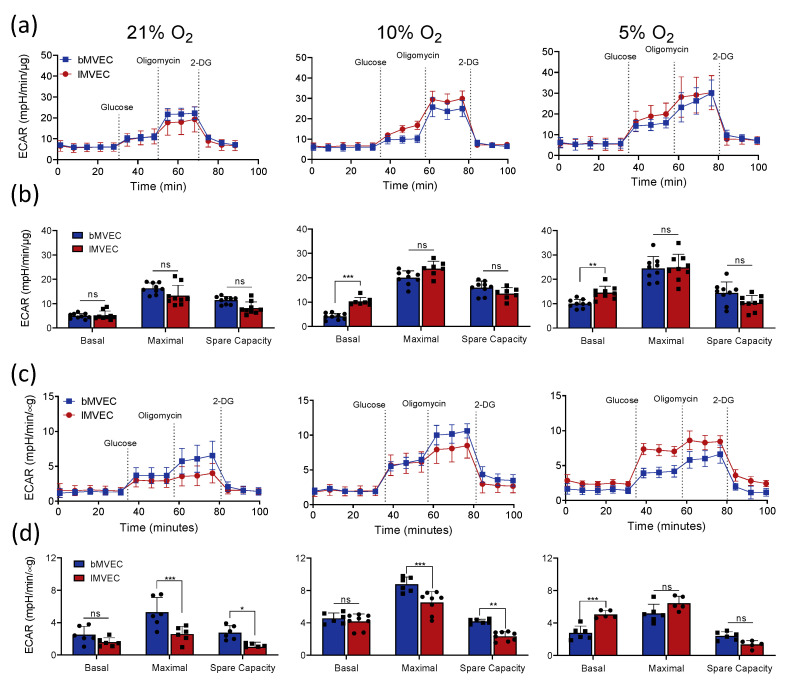
MVEC glycolytic activity is organ-specific and affected by O_2_ priming. (**a**) Representative charts of baseline glycolytic stress tests of brain and lung MVEC cultured at 21% O_2_ (left), 10% O_2_ (middle) or 5% O_2_ (right); (**b**) glycolytic parameters calculated from each curve from (**a**) are shown as average ± SD (*n* ≥ 4). Statistical analysis was carried out using 2-way ANOVA with Holm–Sidak’s multiple comparison test (** *p* < 0.01, *** *p* < 0.001, ns = not significant); (**c**) representative charts of glycolytic stress tests of brain and lung MVEC after 24 h of hypoxia (1% O_2_) from cells primed at different O_2_. Assays were carried out at 1% O_2_; (**d**) glycolytic parameters calculated from each curve from (**c**) are shown as average ± SD (*n* ≥ 4). Statistical analysis was performed using 2-way ANOVA with Holm–Sidak’s multiple comparison test (* *p* < 0.05, ** *p* < 0.01, *** < 0.001, ns = not significant).

**Figure 4 cells-11-02469-f004:**
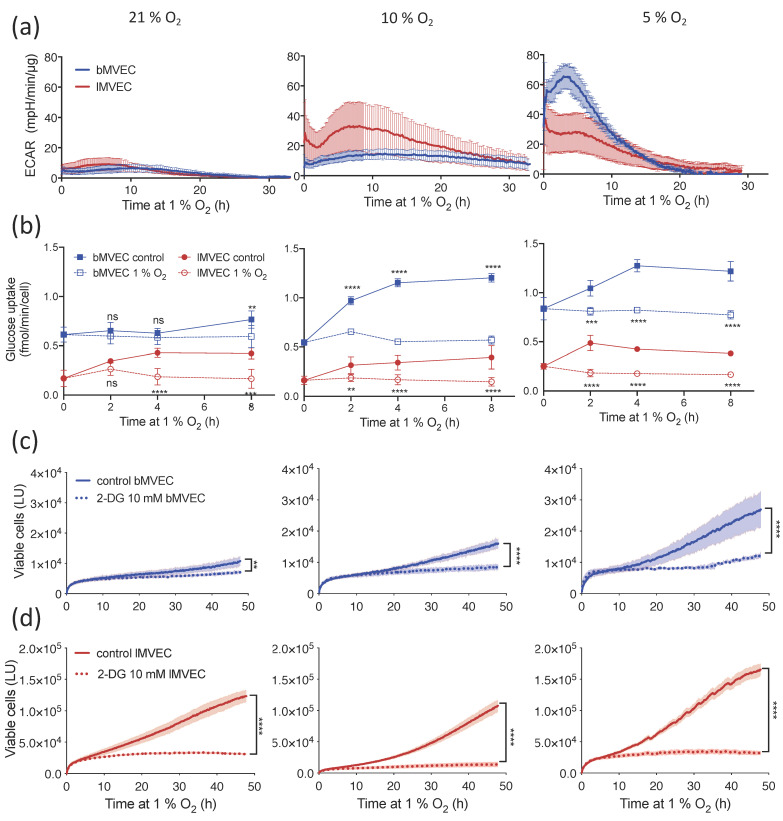
Timing and amplitude of metabolic shift to glycolysis is organ-specific and oxygen-dependent. (**a**) Representative charts of real-time measurement of ECAR changes upon transfer to 1% O_2_ at t = 0, in lung and brain MVEC previously expanded at different O_2_ levels (21% O_2_, left), 10% O_2,_ middle or 5% O_2_, right) (*n* ≥ 4); (**b**) glucose uptake in MVEC expanded at different O_2_ levels at baseline (dashed line) or under hypoxia (solid line), was measured at the indicated times using the Glucose Uptake-Glo Assay (Promega). Data shown as average ± SD, and statistical differences were assessed by 2-way ANOVA with Holm–Sidak’s multiple comparison test (** *p* < 0.01, *** *p* < 0.001, **** *p* < 0.0001), *n* = 3; Real-time viability of (**c**) brain and (**d**) lung MVEC was assessed every 10 min using a Real-Time Glo Assay (Promega) in MVEC transferred to 1% O_2_ at t = 0 and treated with 10 mM 2-deoxy glucose (dotted line) or PBS vehicle control (solid line). *n* = 6 for each condition. Shaded areas show SD, statistical differences were assessed by unpaired Student’s t-test, corrected for multiple comparisons (Holm–Sidak) using the area under the curve (** *p* < 0.01, **** *p* < 0.0001).

**Figure 5 cells-11-02469-f005:**
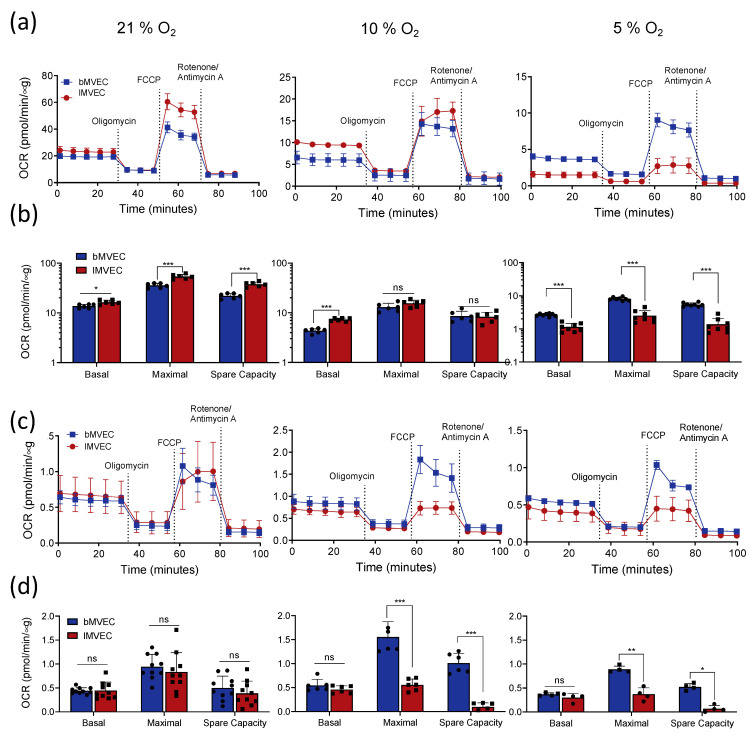
Mitochondrial metabolism of MVEC at baseline and after hypoxia is conditioned by oxygen priming and organ of origin. (**a**) Representative charts of mitochondrial function at baseline, in MVEC cultured at different O_2_ (21%, left, 10% O_2_ middle or 5% O_2_ right); (**b**) mitochondrial metabolic parameters were calculated from each curve from (**a**). Data are shown as average ± SD (*n* ≥ 4), statistical analysis was done using multiple t-tests with Holm–Sidak’s multiple comparison test (* *p* < 0.05, ** *p* < 0.01, *** *p* < 0.001, ns = not significant); (**c**) representative charts of mitochondrial function after adaptation to hypoxia (1% O_2_ for 24 h prior to start of the assay). Assays were carried out at 1% O_2_; (**d**) mitochondrial metabolic parameters of assays carried out after hypoxia were calculated from each curve from (**c**). Data are shown as average ± SD (*n* ≥ 4), Significance assessed with 2-way ANOVA with Holm–Sidak’s multiple comparison test (* *p* < 0.05, ** *p* < 0.01, *** *p* < 0.001, ns = not significant).

**Figure 6 cells-11-02469-f006:**
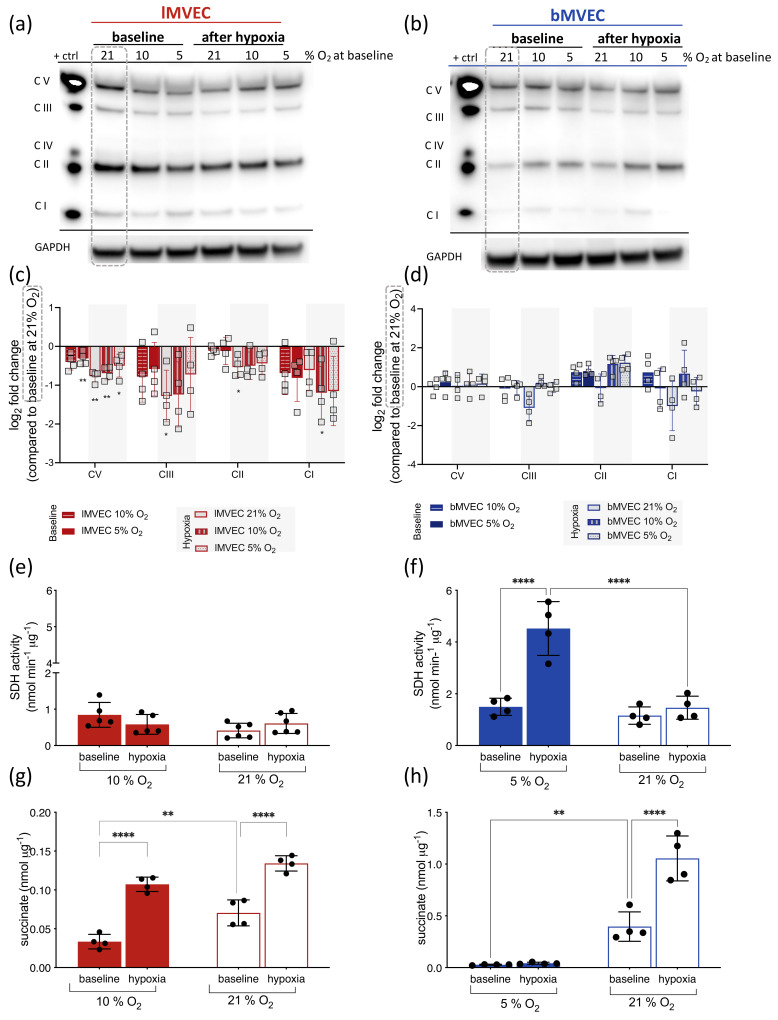
Brain and Lung MVEC have different patterns of mitochondrial ETC components. Whole protein extracts were collected from MVEC from the three baseline (21%, 10% O_2_ and 5% O_2_) and after exposure of each to 1% O_2_ for 24 h. The lysates were probed by Western blot using a mitochondrial antibody cocktail. Representative images are shown for lung (**a**) and brain (**b**) MVEC expanded in each O_2_ environment; (**c**,**d**) MVEC mitochondrial protein quantification (lung and brain, respectively) was performed by densitometry, normalised to loading control (GAPDH) and presented as a ratio to the levels of each mitochondrial complex at 21% O_2_ (dashed grey box in each blot highlighting lane used as “control”); data are shown as log_2_(fold change) ± SD, statistical significance was assessed by one sample *t*-test; * *p* < 0.05, ** *p* < 0.01, *n* = 4. Complex I was detected at levels below threshold and thus could not be accurately quantified. SDH activity for *l*MVEC (**e**) and *b*MVEC (**f**) and succinate levels (**g**,**h**) for lung and brain MVEC, respectively) were measured at baseline and after 24 h hypoxia 1% O_2_), following priming in either physioxia (5% O_2_ for brain, 10 % O_2_ for lung) or hyperoxia (21% O_2_). Data are shown as average ± SEM and significance was assessed by 2-way ANOVA, Holm–Sidak’s multiple comparison test, ** *p* < 0.01, **** *p* < 0.0001, *n* = 4.

## Data Availability

All data generated or analysed during this study are included in this published article and its Appendix A. Further information and requests for resources and reagents should be directed to and will be fulfilled by the corresponding author, Cristina M Branco (c.branco@qub.ac.uk) on reasonable request. This study did not generate new unique materials.

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
