# Peer review of "Hyperoxia Reprogrammes Microvascular Endothelial Cell Response to Hypoxia in an Organ-Specific Manner"

_cells, 2022, doi:10.3390/cells11162469_

Round 1
Reviewer 1 Report
The present study evaluated the effect of different oxygen tension on metabolism of microvascular endothelial cells and its response to hypoxia. Briefly, the authors have shown that exposure to high oxygen concentrations can decrease cell viability and impairs mitochondrial function. Besides, high oxygen exposure impairs the metabolic response of endothelial cells in conditions of hypoxia. Such results are very interesting and relevant for studies evaluating endothelial cell metabolism, in particular those focused on ischemia-reperfusion investigation. Overall, the manuscript is well written and the experimental design is appropriate, the data is novel and can bring important contributions for the field. I just have few comments/suggestion:
1) Which passage were the cells used for experiments? Please, add this information.
2) On Figure 1C, loading control is described as TATA-BP instead of actin, like Figure 1D. Is it correct or it’s a typo? Why did the authors choose it?
3) Please, add the molecular weight of the ladders to the figures of the whole membranes.
4) On Figure 2, the graph g, which follows f, is marked as e. Please, amend it and add a legend for the bar graphs.
5) The description of results for Figure 6 is different of the legend, eg Fig 6c is described as WB of bMVEC but, in fact, it is a graph showing the measurements of immunoblots for lMVEC. Please amend it accordingly.
Reviewer 2 Report
This study investigates the previously unexplored role of hyperoxia on vascular bed-specific responses to hypoxia in lung and brain microvascular cells. Using cell-based metabolic and functional assays, the authors defined how hypoxic pre-exposure can modulate specific endothelial metabolic (glycolysis and OXPHOS), angiogenic, and growth responses. The manuscript is well written and provides new information in the field of vascular adaptation to hypoxia. However, some suggestions would strengthen the manuscript.
1. The manuscript would be deeper if the authors analyze a set of additional transcription factors in bMVEC and lMVEC in response to hyperoxic reprogramming.
2. Correct some conclusions. For example, the statement on line 304 says that hypoxic LMVEC increasingly accumulated VEGF, whereas the data in Figure 1e do not show any dramatic effect.
Missed data on bMVEC ARG2 expression (48 hrs) should be included.
3. Inhibition of cell viability by 2-DG suggests the dependence not only on glycolysis. The resulting decrease in pyruvate production may also suggest dependence on OXPHOS.
4. Manuscript length. I would suggest condensing “Materials and Methods” by including references to previously published methods. In addition, the “Results” section should be more succinct.
5. “Materials and Methods” section 2.10: Double-check and correct the concentrations of Oligomycin, 2-Deoxy glycose, Antimycin, and Rotenone. Are they in the millimolar range?
6. Statistical analysis. It is not clear why the authors compare brain MVEC vs Lung MVEC (figures 1 and 3 ), but not hypoxic responses in each cell type vs basal conditions
7. Explain TATA-BP in the Figure 1c
8. Gramma corrections (lines 281 and 391): use plural verbs for “data”
9. Format references (lines 410-411)
Round 2
Reviewer 2 Report
The authors did a good job to address all the comments and suggestions